# Symptoms of Attention Deficit/Hyperactivity Disorder Are Associated with Sub-Optimal and Inconsistent Temporal Decision Making

**DOI:** 10.3390/brainsci12101312

**Published:** 2022-09-28

**Authors:** Ortal Gabrieli-Seri, Eyal Ert, Yehuda Pollak

**Affiliations:** 1The Seymour Fox School of Education, The Hebrew University of Jerusalem, Jerusalem 9190501, Israel; 2Department of Environmental Economics and Management, The Hebrew University of Jerusalem, Rehovot 7610001, Israel

**Keywords:** ADHD, decision-making, delay discounting, suboptimal, consistency

## Abstract

The link between Attention-Deficit/Hyperactivity Disorder (ADHD) and steeper delay discounting has been established and incorporated into theories of ADHD. This study examines a novel interpretation according to which ADHD is linked to sub-optimal temporal decision-making and suggests inconsistency as a potential underlying mechanism. In two experiments, MTurk workers completed a self-report questionnaire on symptoms of ADHD and a temporal decision making task consisting of choices between smaller–immediate and larger–delayed options. The delayed option was better in some items, whereas the immediate option was better in others. The rate of choices of the delayed option and the consistency of choices were measured. The results of both studies show that high symptoms of ADHD were linked to fewer choices of the delayed option when it was better, but also to more choices of the delayed option when it was not better. In addition, ADHD was linked to higher inconsistency in both conditions. The findings suggest that ADHD is linked to sub-optimal temporal decision-making rather than steeper delay discounting, and provide further support to the phenomenon of inconsistency in ADHD.

## 1. Introduction

Attention-Deficit/Hyperactivity Disorder (ADHD) is a neuro-developmental condition defined by persistent symptoms of inattention and/or hyperactivity–impulsivity interfering with functioning or development [1]. ADHD is highly prevalent (5.9% in youth and 2.5% in adults) and has been proven to reduce various dimensions of quality of life [2,3,4,5]. A significant deficit that has been related to ADHD is impaired time-related (temporal) decision-making [6,7,8], which was thought to reflect impulsivity, a core symptom of ADHD [9,10].

Time-related decisions are traditionally characterized by the term delay discounting, meaning that a future payoff has a lower utility than an equivalent payoff in the present. The research on this topic is vast; however, it is mostly agreed that delay discounting is subjective and can be determined experimentally (for a review, see Frederick et al. [11]). For example, Mazur [12] presented a discounting rate parameter *k* (in the function $V=\frac{A}{1+kD}$, where *V* is the immediate reward, *A* is the delayed reward, and *D* is the length of the delay) that is computed from the participant’s pattern of choice in a monetary choice questionnaire [13].

Research on temporal decisions often includes a temporal discounting task: the participant is asked to choose between two monetary rewards, a small and immediate or a larger but delayed [9,14]. As mentioned above, regarding ADHD, the evidence about the link between the disorder and delay discounting supposedly points to a bias towards steeper delay discounting [6], i.e., an even lower utility assigned to future payoff than that in controls. Specifically, it has been found that people with ADHD tend to choose the small immediate reward over the larger delayed reward more often than controls [6]. The findings have often been interpreted as reflecting steeper delay discounting modeled by a larger discount parameter among people with ADHD.

While the evidence for immediacy bias seems clear, the studies documenting this tendency have focused on cases where the delayed reward option is better than the immediate one. This focus is well justified and characterizes many real-life temporal dilemmas. However, there might be situations in which it is better to choose the immediate payoff, or at least most people would agree. For an obvious example, consider many banks saving accounts that currently offer incredibly low interest rates. Most people would agree that investing USD 1000 in an account with a yearly interest of 0.01% is not worthwhile or profitable. The calculated *k* parameter for this problem (i.e., USD 1000 now or USD 1010 after one year) is around 0.00003, which is very low, so an individual’s *k* value needs to be at this level or lower for them to accept such an offer. Despite such examples, studies of time preferences of people with ADHD have implicitly assumed that the delayed reward is the better choice. In other words, optimality and delay have been confounded, which raises the question of whether this confounding might have led to overlooking alternative explanations for the immediacy tendency among people with ADHD.

An alternative explanation to the temporal decision-making deficit in ADHD is a sub-optimal valuation of delay. This novel interpretation states that ADHD might be related to sub-optimal choices regardless of immediacy. Specifically, when the delayed option is better, people with ADHD might indeed lean towards the immediate option. Still, when the immediate option offers the better choice, people with ADHD might lean towards the less attractive delayed option. This latter tendency, if it exists, contradicts the previously mentioned conclusions regarding steeper delay discounting. Specifically, if ADHD is linked with immediacy preference, people with ADHD should have an advantage in immediate-is-better situations. The first goal of this research is to compare these distinct interpretations.

The current study’s second goal is to explore choice inconsistency as a possible explanation for temporal decision-making in ADHD. The consistency of preferences is a core element for making optimal decisions [15], and inconsistency in decision-making might lead to sub-optimal decisions, [16]. ADHD has been linked to inconsistency in different aspects, such as reaction times [17,18,19], time estimation [20], working memory [21], driving behavior [22], and self-reports of delinquency and impulsivity [23,24]. In the context of decision-making, ADHD has been related to inconsistency in decisions under risk, and was suggested as a possible explanation for sub-optimal decision-making [25]. The current study tests whether ADHD is linked to higher inconsistency in temporal decision-making.

The current paper examined the sub-optimal valuation and inconsistency hypotheses using two studies. In Experiment 1, we used a between-subject design to explore the hypotheses that people with a high level of symptoms of ADHD will show sub-optimal decision-making in both “better–delayed” and “better–immediate” conditions. Experiment 2 used a within-subject design, where the same participants chose between two options under both better–delayed and better–immediate conditions.

## 2. Materials and Methods

### 2.1. Experiment 1

#### 2.1.1. Participants

The sample consisted of 200 Amazon Mechanical Turk workers (age range = 18–74, 49.5% female) with a “Master” qualification (workers with high credibility). The participants received a compensation of USD 0.8 for their participation. After giving their informed consent, participants were randomly assigned to the study task conditions. All studies reported here were approved by the Ethics Committee of the Seymour Fox School of Education, The Hebrew University of Jerusalem.

#### 2.1.2. Materials

A revised version of the monetary-choice questionnaire (MCQ) based on Kirby et al.’s [13] scale assessed the delay discounting level. Participants were presented with a fixed set of choices between immediate rewards and larger delayed rewards, e.g., “Would you prefer $30 today or $35 in 80 days?” and marked their preferred option. The scales included 27 questions (See Table A1). The order of questions was contrived so that the trial order would not correlate with the immediate or delayed amounts, their ratio, their difference, the delay to the larger reward, or the discount rate (*k*) corresponding to indifference between the two rewards. The original scale was split, and items were added to create two scales of 27 items, one with low *k* values (0.00016–0.0025) and one with high *k* values (0.045–0.35), to create the two conditions “better–immediate” and “better–delayed”. The items were presented in a fixed order to ensure that the *k* values were mixed properly (following Kirby et al. [13]).

The Adult ADHD Self Report Scale (ASRS-V1.1) [26], adapted for computer presentation, was completed for continuous scaling of ADHD symptoms. A dimensional model of ADHD was adopted for this study, as taxometric and genetic evidence has shown that a dimensional conceptualization of ADHD has merits for research purposes [27]. The ASRS-V1.1 contains 18 items corresponding to the DSM-IV diagnostic criteria for ADHD. The symptoms are rated on a Likert 5-point scale by frequency of occurrence ranging from “never” to “very often”. The questionnaire has high validity and reliability (internal consistency, α = 0.88) in assessing ADHD in adults. Its reported sensitivity is 68.4% and its specificity is 99.6% [28]. The ASRS score was calculated by averaging the scores of all 18 items (1 = low, 5 = high).

Attention check questions. To verify that the participants attended to the content of the task, two questions were added: in the middle of the MCQ task, the participants were asked to choose the immediate option (“If you read this, please choose $33 today”), and in the middle of the ASRS questionnaire, to select “often” (“If you read this, please choose “often”).

In addition, participants reported demographic information (age, gender, income level, and years of education) and ADHD-related information (ADHD medication on the day of the experiment and a history of ADHD diagnosis).

#### 2.1.3. Data analysis

In total, 2 participants who failed the attention check and 23 who completed the whole study in less than four minutes were excluded from further analysis (all reported effects remained when the analysis included all 198 participants, except for the immediate condition, in which the effect for the proportion of delayed options did not reach significance). The latter criterion was applied to improve data quality, which was found to be poorer when “speeders” were included in online samples [29]. The final sample consisted of 175 participants (see Table 1 for descriptive statistics by optimality condition).

The consistency score was calculated by ordering the items and computing the percentage of choices most consistent with an indifference point at one of the different *k* values of the questionnaire [13]. For example, suppose a participant chose the immediate option in the first 24 items (ordered by discounting rate) and switched to the delayed option for the last 3 items with the higher rate. In that case, their choices were 100% consistent with the *k* value of the 24th item. However, suppose a participant mainly chose the immediate option and only chose the delayed option at the 6th, 8th, 10th, 25th, and 26th items. In that case, their assigned *k* value will be identical to the previous participant, but with a consistency score of 85%. The discounting rate was operationalized as the percent of times a participant chose the delayed reward. The variables age, gender, socioeconomic position, and education were controlled. Regression analyses were performed to examine the relationship between ADHD, delay discounting, and optimality. Considering previous findings of the relationship between delay discounting and psychiatric variables [30], we examined a quadratic model in addition to the linear one. Specifically, when it is better to choose the delayed reward, we expected a higher rate of choosing the immediate option, with high levels of ADHD symptoms. In the better–immediate condition, we expected the opposite: higher delay scores for high symptoms of ADHD.

### 2.2. Experiment 2

Experiment 2 was conducted to evaluate the robustness of Experiment 1′s findings using a within-subject design. The same participants chose between two options under both better–delayed and better–immediate conditions, enabling us to examine whether the same participants made sub-optimal choices under both conditions.

#### 2.2.1. Participants

One hundred participants (age range = 18–74 (*Mdn* = 25–34); 31% female) were recruited the same way as described in Experiment 1. The participants were instructed to choose the option they preferred in each of the 27 choice problems presented. The data of 14 participants were removed from the analysis due to response times shorter than four minutes.

#### 2.2.2. Materials

The experiment used the same tasks and materials as in Experiment 1, with the following adaptations of the main task for a within-subjects design:

The monetary-choice questionnaire (MCQ) based on Kirby et al.’s [13] scale was revised to include more values of *k*, and the same scale of 27 items (see Table A2) was presented to all participants. The scale included 18 items corresponding to Experiment 1’s “better–delayed” and “better–immediate” conditions (9 each); the remaining 9 items had “medium” *k* values, intended to mask the large difference between the low and high items.

Data analysis. The questionnaires and the task were coded the same way as in Experiment 1. A repeated-measures ANOVA was used with low, medium, and high *k* levels of the 27 choices as the within-subjects variable. The ASRS score and ASRS squared score were continuous covariates. A second analysis used consistency score as the within-subjects variable. Demographics and clinical information of the participants are presented in Table 2.

## 3. Results

### 3.1. Experiment 1

Regression analyses were used to examine linear and curvilinear relations between the level of symptoms of ADHD and sub-optimal temporal decision-making. Separate analyses were performed for the two conditions (“better–delayed” and “better–immediate”). In the better–delayed condition, the linear model was significant (*F* (1, 91) = 30.10, *p* < 0.001, *R*^2^ = 0.25), explaining 25% of the variance in delayed choices. The results reveal that people with higher ASRS scores chose more immediate rewards than those with lower scores (*β* = −0.50, *p* < 0.001, 95% CI (−0.68, −0.32)). The quadratic model had better fit (*F* (2, 90) = 19.60, *p* < 0.001, *R*^2^ = 0.30), showing that the level of delayed choices increased and then (more steeply) dropped with the increase in ASRS scores (*β* = −1.41, *p* = 0.009, 95% CI (−2.46, −0.36)). In the better–immediate condition, the linear model was insignificant (*F* (1, 80) = 0.57, *p* = 0.452, *R*^2^ = 0.08), while the quadratic model was significant (*F* (2, 79) = 4.32, *p* = 0.017, *R*^2^ = 0.10), explaining 10% of the variance in delayed choices. It showed that ASRS significantly predicted delayed choices in an opposite pattern from the “better–delayed” condition (*β* = 1.68, *p* = 0.006, 95% CI (0.50, 2.86)). That is, the pattern of the quadratic model shows that the level of delayed choices decreased and then increased with the rise in ASRS score. The quadratic curves in both conditions are depicted in Figure 1a.

We next examined whether the level of ADHD affects choice consistency. Specifically, we regressed the rate of consistent choices on the ASRS score under each of the two aforementioned regression models (see Figure 1b for the quadratic curves). In the better–delayed condition, the linear model was significant (*F* (1, 91) = 51.59, *p* < 0.001), explaining 36% of the change in delayed choices (*R*^2^ = 0.36). The quadratic model improved the prediction to 44% (*F* (2, 90) = 35.38, *p* < 0.001, *R*^2^ = 0.44). In the better–immediate condition, the linear model was also significant (*F* (1, 80) = 39.11, *p* < 0.001), explaining 33% of the change in delayed choices (*R*^2^
*=* 0.33). The quadratic model improved the prediction to 43% (*F* (2, 79) = 29.74, *p* < 0.001, *R*^2^ = 0.43). In both conditions, a higher ASRS score significantly predicted less consistency in choices (*β* = −0.60, *p* < 0.001, 95% CI (−0.77, −0.43); *β* = −0.57, *p* < 0.001, 95% CI (−0.75, −0.39); respectively). According to the quadratic model, the level of consistency somewhat increased and then (more steeply) dropped with the rise in ASRS scores, in both conditions (*β* = −1.68, *p* = 0.001, 95% CI (−2.63, −0.74); *β* = −1.76, *p* < 0.001, 95% CI (−2.70, −0.83); respectively).

The results suggest that participants who reported a high level of ADHD symptoms showed lower consistency than participants with a low level of ADHD symptoms, independently of the optimality conditions. Participants who reported a high level of ADHD symptoms showed lower consistency than participants with a low level of ADHD symptoms, independently of the optimality conditions.

When the delayed option was better, participants with a high level of ADHD chose the immediate option more often than participants with a low level of ADHD, replicating the pattern of delay discounting as documented by previous research. Yet when the immediate option was better, participants with high levels of ADHD symptoms chose the delayed option more often, in contrast to the steeper delay discounting hypothesis. Furthermore, choices made by the high ADHD participants were less consistent under both conditions. The results suggest that people with a high level of ADHD symptoms do not always exhibit a stronger preference for immediate rewards, as assumed by the literature, but might instead exhibit less optimal temporal decision-making than controls.

### 3.2. Experiment 2

We conducted a repeated-measures ANOVA with the three levels of *k* (low, medium, high) as the dependent variables, and the ASRS score and ASRS squared score as continuous covariates, to examine the hypothesis of sub-optimal decision-making in relation to high levels of symptoms of ADHD. The analysis yielded a significant interaction effect between both covariates and delayed choices (ASRS: (*F* (2, 82) = 6.43, *p* = 0.003, $\eta_{p}^{2}=0$.14, 95% CI (0.20, 0.26)); ASRS squared: (*F* (2, 82) = 10.11, *p* < 0.001, $\eta_{p}^{2}=0$.20, 95% CI (0.05, 0.330)). The main effect was not found for delayed choices (*F* (2, 82) = 0.67, *p* = 0.52).

To explore the delay × ASRS interaction, we performed separate post hoc regression tests for the three *k* levels (Figure 2a). When the *k* level was high, i.e., the delayed reward was better, the linear model was significant (*F* (1, 84) = 27.47, *p* < 0.001, *R*^2^ = 0.25), explaining 25% of the variance in delayed choices. ASRS significantly predicted less delayed choices (*β* = −0.50, *p* < 0.001, 95% CI (−0.70, −0.31)). The quadratic model improved the prediction to 30% (*F* (2, 83) = 17.97, *p* < 0.001, *R*^2^ = 0.30), which implies a rise in choices of the delayed reward that dropped with the rise in ASRS score (*β* = −1.26, *p* = 0.012, 95% CI (−2.30, −0.29)). When the *k* level was low, and it was better to choose the immediate reward, the linear model was insignificant (*F* (1, 84) = 0.31, *p* = 0.580). However, the quadratic model was significant and explained 10% of the variance (*F* (2, 83) = 4.78, *p* = 0.011, *R*^2^ = 0.10), according to which there was a decrease in the delay options chosen and then a rise compliant with the rise in ASRS score (*β* = 1.68, *p* = 0.003, 95% CI (0.09, 0.42)). For the middle items, the results were similar to those of the high-level delay items.

To test the hypothesis of inconsistency related to a high level of symptoms of ADHD, we performed a second repeated measures ANOVA with consistency scores of the previous *k* levels (low, medium, high) as the dependent variable. The analysis yielded a significant main effect for ASRS and the quadratic covariate ASRS squared (*F* (1, 83) = 13.75, *p* < 0.001, $\eta_{p}^{2}=0$.14, 95% CI (0.03, 0.28); *F* (1, 83) = 33.10, *p* < 0.001, $\eta_{p}^{2}=0$.29, 95% CI (0.13, 0.42); respectively), indicating that higher ASRS levels were related to lower consistency levels (Figure 2b). Interaction effects between both covariates and consistency scores were not found (all *p*’s > 0.05).

The results show that participants with a high level of ADHD symptoms demonstrated sub-optimal choices both when the better choice was the immediate one and when it was the delayed one. In other words, the same person with a high level of ADHD symptoms was more likely to choose both the immediate option when the delayed was better and the delayed option when the immediate was better. Further, high ADHD was related to less consistency in both contexts. The results support the claim that ADHD is characterized by sub-optimal and inconsistent temporal decision-making rather than steeper delay discounting and an immediacy bias.

## 4. Discussion

Previous studies have suggested that people with ADHD are characterized by a higher immediacy bias than controls in their temporal decisions [6,7]. Yet these studies did not differentiate between settings where the delayed option is better and those where the immediate option is better. This study explored an alternative explanation for ADHD temporal decisions, suggesting that people with ADHD might be characterized by sub-optimal decisions and are less consistent, rather than being merely impatient. An intriguing implication of this mechanism is that people with ADHD might show more patience than controls in situations where they actually should choose the immediate option.

Two experiments were conducted to confront these competing hypotheses. The first compared a delay discounting paradigm where the delayed option was better with a condition where the immediate option was better. The second experiment tested whether the results of the first experiment were replicated at the individual level using a within-subject design. The results support the hypothesis that a high level of ADHD symptoms is related to sub-optimality in the valuation of the delay, rather than steeper delay discounting. This pattern implies that a person with ADHD might appear either impatient when the delayed reward is better or too patient when the immediate is better.

Sub-optimal decision-making in people with ADHD was also evident in research on decision-making under risk. Dekkers et al. [31] conducted a study where they included risky items with both high and low expected values (i.e., sometimes it was better to choose the risky gamble, and sometimes it was better to choose the safe option). They found that people with ADHD chose the risky option more often than controls when it was associated with a low expected value, but chose the risky option less often than controls when it was associated with a high expected value. These findings support the sub-optimal hypothesis rather than the risk-seeking or risk-taking bias hypothesis. Further research is needed to evaluate the possibility that sub-optimal decision-making by people with ADHD generalizes to other domains beyond risk and temporal decisions.

The second aim of this study was to explore inconsistency as a possible explanation for sub-optimal decision-making in ADHD. How can inconsistency explain the sub-optimal decisions of people with ADHD? For an example of a temporal discounting decision, the choice of whether to wait is determined by the discounting rate assigned to the delayed reward. For example, when facing a series of trials in which the immediate option is better (i.e., the *k* value is relatively high), typically, the discounting rate of the participant would be relatively consistent across trials, resulting in choosing the better option for most trials. A participant with a high level of ADHD would have a similar discounting rate on average. However, due to inconsistency, in some trials, the discounting rate would be lower than that of the participant with low ADHD, resulting in choosing the immediate option. Nevertheless, in other trials, the discounting rate of the participant with high ADHD would be higher than that of the participant with low ADHD, resulting in more choices of the delayed reward, which is the worse option in this example. As such, this inconsistency can account for sub-optimal choices, regardless of their direction.

It is worthwhile noting that the current study does not rule out the possibility that ADHD is related to steeper delay discounting. It simply shows that steeper delay discounting cannot account for all ADHD-related temporal decisions, and specifically in situations where the immediate option is better. Therefore, steeper delay discounting is not the only explanation for the temporal choices of people with ADHD, and the inconsistency that characterizes their decisions explains at least part of the decision-making process in ADHD. Further, decision-making inconsistency in relation to ADHD was also found in the context of risky decision-making in a subsequent analysis of the data from Dekkers et al. [31]. Specifically, ADHD was associated with lower consistency in the weight given to the risk [25]. Evidently, inconsistency in ADHD is a strong component of both conditions presented here, and we suggest that it is the main driver behind the sub-optimal decision-making of people with ADHD.

We tested both linear and quadratic models. In all comparisons, the quadratic model was superior, suggesting that ADHD does not affect temporal decision-making monotonously as far as decision-making is concerned. Much like other cognitive, personality, and psychiatric qualities, it is probably true that any extreme is not ideal [32]. This tendency may open up new directions for research into other ADHD-related behaviors.

Every study has limitations, and this one is not exceptional. First, this study’s definition of high and low ADHD was based on the ASRS questionnaire that probes symptoms only. It is possible that recruiting a clinically diagnosed ADHD group could have shed some light on the role of functional impairment in the link between ADHD and sub-optimal decision-making, which was neglected in the current investigation. Further, it is possible that the use of the ASRS, valid and reliable as it is, might not yield clinical implications, because it may indicate a more general cognitive dysfunction that is not necessarily ADHD. However, we based our choice on the support for a dimensional perspective, rather than a categorical one, on ADHD [27].

Second, old age was not considered an exclusion criterion, and therefore the age range was wide (18–74). Importantly, our findings could hence be attributed to the advanced age of some of our participants, and the related neurological diseases and deficits. However, re-examining the data with age as a covariate did not have an impact on the results. We further made sure that participants did not report taking any medication for the treatment of neurological diseases. Future studies should take that issue into account in advance.

Third, this study explored delay discounting using an experimental task. The use of the present and similar tasks to examine delay discounting is accepted and well-spread [6,7,8]. Yet, these tasks might be criticized for not representing real-world decisions that may involve delay discounting. That said, the scope of the current study was limited to the, mainly, theoretical understanding of the link between ADHD and temporal decision-making, and for that, an experimental task was sufficient.

Fourth, some authors recommended excluding responses with consistency scores lower than 0.8 in the MCQ paradigm [33]. Here, we hypothesized that inconsistency would be related to ADHD, and therefore, we did not implement that exclusion criterion. However, we repeated our analysis with the exclusion criterion to test whether our findings would emerge even with consistency scores only above 80%. The results show that the effects of the condition where the delayed options were better remained significant, but the weaker effects of the better–immediate condition did not reach significance. An examination of the ADHD scores (ASRS mean) before and after the filtering revealed that the original mean of 2.36 (SD = 0.73) had dropped to 2.16 (SD = 0.55). On the other hand, the ADHD level of the 53 filtered participants was significantly higher (M = 2.83, SD = 0.87; t(196) = −5.89, *p* < 0.001). Therefore, it seems that employing the exclusion criterion removed many of our high-ADHD participants, preventing the replication of some of the findings. Importantly, this demonstration suggests that consistency, or inconsistency, is a fundamental mechanism of ADHD. It also indicates that inconsistency appears together with steeper delay discounting when the delayed option is optimal. However, when the immediate option is optimal, a high inconsistency that characterizes the far end of the ADHD symptoms continuum may be the source of the effects we reported in the results section above.

Lastly, the suggested mechanism of inconsistency is not the only plausible explanation. In fact, it has been suggested that the sub-optimal decision-making of people with ADHD can be caused by the use of less complex strategies [34]. Further research should explore causality, confront the different explanations, or coherently unify them.

Further research should also investigate sub-optimality in the decision-making process of people with ADHD from a developmental perspective [35], with clinical samples, and study the link in real-life delay situations. Future studies should also focus on other domains of decision-making, such as social, financial, and health-related decision-making. They should also examine whether inconsistency in decision-making relates to inconsistency in performance, e.g., variability in reaction time on cognitive tests. Importantly, future research should further develop inconsistency as a theoretical explanation of the phenomena, and examine the option to utilize that construct for clinical practice.

## 5. Conclusions

In conclusion, the current study’s findings suggest that ADHD symptoms might not be linked to steeper delay discounting, but rather to a sub-optimal valuation of the delay, leading to either impatient or overly patient sub-optimal choices. Importantly, ADHD is linked to general inconsistency, which can account for the sub-optimal decision-making deficit demonstrated in this study. Further research should investigate sub-optimality in the decision-making of people with ADHD, and inconsistency as a mechanism.

## Figures and Tables

**Figure 1 brainsci-12-01312-f001:**
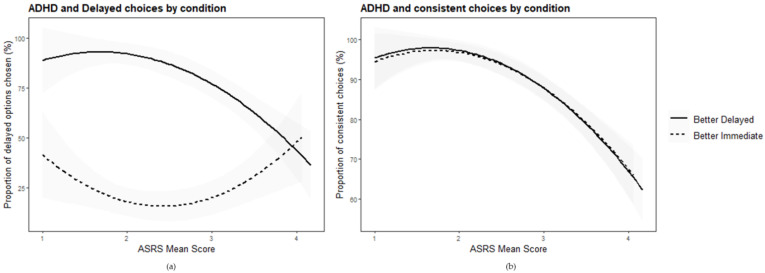
(**a**) Regressions (quadratic term) of the relationship between the level of ADHD symptoms (ASRS (ADHD Self Report Scale) mean score) and the proportion of choices to wait for the two conditions: delay is better (solid line), and immediate is better (dashed line; *R*^2^ = 0.30; *R*^2^ = 0.10, respectively). (**b**) Regressions (quadratic term) of the relationship between the level of ADHD symptoms (ASRS mean score) and the proportion of consistent choices for the two conditions: delay is better (solid line), and immediate is better (dashed line; *R*^2^ = 0.44; *R*^2^ = 0.43, respectively).

**Figure 2 brainsci-12-01312-f002:**
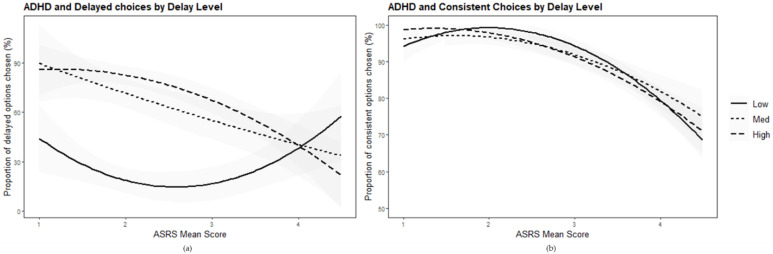
(**a**) Regressions (quadratic term) of the relationship between the level of ADHD symptoms (ASRS mean score) and the proportion of choices to wait for low (solid line), medium (dotted line), and high (dashed line) levels. i.e., low = better immediate and high = better delay (*R*^2^ = 0.11; *R*^2^ = 0.31, respectively). (**b**) The main effect of ADHD (quadratic) on the proportion of consistency in the three levels: low (solid line), medium (dotted line), and high (dashed line).

**Table 1 brainsci-12-01312-t001:** Demographic and clinical characteristics by optimality condition groups.

	Immediate Is Better (*n* = 82)	Delayed Is Better (*n* = 93)	Group Comparison
Age range Mode (%)	35–44 (37.8%)	35–44 (33.3%)	*t*(169) = 1.71 (*p* = 0.088)
Gender ratio	37 females (45.1%)	49 females (52.7%)	χ^2^(1) = 0.72 (*p =* 0.397)
Annual Income category Mode (%)	USD 10,000–24,999 (28%)	USD 10,000–24,999 (22.6%)	*χ*^2^(7) *=* 9.21 (*p =* 0.243)
Education level Mode (%)	Undergraduate (42.7%)	Undergraduate (33.3%)	*χ*^2^(4) *=* 4.77 (*p =* 0.314)
Self-reported history of ADHD diagnosis	Three diagnosed (3.7%)	Six diagnosed (6.5%)	*χ*^2^(1) = 0.24 (*p =* 0.504)

**Table 2 brainsci-12-01312-t002:** Demographic and clinical characteristics, Experiment 2.

	*n* = 91
Age Mean (SD)	42.92 (11.32)
Gender ratio	37 females (41%)
Annual income category Mode (%)	USD 25,000–49,999 (26%)
Education level Mode (%)	Undergraduate (40%)
Self-reported history of ADHD diagnosis	Five diagnosed (5.5%)

## Data Availability

The data presented in this study are available upon request from the corresponding author.

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
