# Peer review of "Symptoms of Attention Deficit/Hyperactivity Disorder Are Associated with Sub-Optimal and Inconsistent Temporal Decision Making"

_brainsci, 2022, doi:10.3390/brainsci12101312_

Round 1
Reviewer 1 Report
This is an excellently designed and well written study on an important topic for research and clinic. The issue of "delay discounting" in ADHD is often simplified in text books and media reports, and even in research reviews, so I appreciate this effort of the authors to give a more nuanced perspective and add evidence to this complex topic. The consideration of sub-optimal temporal decisions, and inconsistent decision making, gives valuable insights into cognitive functioning of adult ADHD and may contribute to our understanding of ADHD. Inconsistency is a phenomenon that is observed commonly in ADHD, in various processes and functions, and thus matches our understanding and knowledge about ADHD.
As a limitation, this study did not include individuals diagnosed with ADHD (please bear in mind that an ADHD questionnaire may indicate general cognitive dysfunctioning or pathology, but is not specific for ADHD), so the clinical implications should be tempered. Further, delay discounting tasks are criticized for their artificial and experimental set-up, so one has to be careful to conclude on real life decision making.
Author Response
We are thankful for your positive impression. Following your important comment, we have significantly increased the limitations section in the discussion. Specifically, paragraphs 7 and 9 of the discussion include the suggested limitations: "…. Further, it is possible that the use of the ASRS, valid and reliable as it is, might not yield clinical implications because it may indicate a more general cognitive dysfunction that is not necessarily ADHD…";", this study explored delay discounting using an experimental task. The use of the present and similar tasks to examine delay discounting is accepted and well-spread [6-8]. Yet, these tasks might be criticized for not representing real-world decisions that may involve delay discounting. That said, the scope of the current study was limited to the, mainly, theoretical understanding of the link between ADHD and temporal decision making, and for that, an experimental task was sufficient."
We also added a suggestion to examine realistic delay discounting in future studies.
Reviewer 2 Report
Thank you for the opportunity to review this interesting article entitled "ADHD is associated with sub-optimal and inconsistent temporal decision making".
First of all, I consider that english version of this manuscript must be checked.
In the title, please do not use abbreviations.
Introduction
-Can you provide epidemiological data in the first paragraph?
-I would delete "this study explores a different explanation".
-"For a review, see [7]" I consider that this style is not appropiated. Please clarify.
-Why in this article appears two studies? It would be two experiments?
-Really, is better a general discussion that two mini-discussions.
-Please, if are present, increase the limitations.
-I consider that the number of references is poor.
Author Response
Reviewer #2
Comment 1: First of all, I consider that english version of this manuscript must be checked.
Response 1: Thank you for this suggestion. The revised manuscript was sent to an English editor to check before resubmission.
Comment 2: In the title, please do not use abbreviations.
Response 2: We removed the abbreviations from the title.
Comment 3: Introduction
-Can you provide epidemiological data in the first paragraph?
Response 3: Epidemiological data on ADHD prevalence was added to the first paragraph: "5.9% in youth and 2.5% in adults" with the appropriate reference.
Comment 4: I would delete "this study explores a different explanation".
Response 4: We are thankful for the suggestion; the sentence was deleted.
Comment 5: "For a review, see [7]" I consider that this style is not appropiated. Please clarify.
Response 5: Corrected, thank you. We also reviewed again the paper to ensure that it is clean from similar mistakes.
Comments 6-7: Why in this article appears two studies? It would be two experiments?
-Really, is better a general discussion that two mini-discussions.
Response 6-7: Following your suggestions, we changed the headings and wording accordingly to include two experiments in the methods and results sections and one general discussion.
Comment 8: -Please, if are present, increase the limitations.
Response 8: We increased the limitations in the discussion section (paragraphs 7-11).
Comment 9: -I consider that the number of references is poor.
Response 9: We added a few relevant papers (specifically, references 3-5, 8, 17, 21, 24, and 28 were added).
Comment 10: I found confusing the use of ‘expected’ in the equation, whereas in the text the word ‘perception’ is more often used… it seems best to use the word ‘perceived’ throughout.
Response10: There appears to be some confusion. The manuscript does not include the word 'expected' in the equation, and we did not use the word 'perception' throughout the manuscript.
Reviewer 3 Report
Dear Authors;
I reviewed the article. Although it is an interesting subject, I think it has important gaps.
My suggestions are as follows.
- In the discussion, it was stated as a limitation that ADHD could not be diagnosed clinically. However, the title is still very specious. Changing the title is recommended as ‘’Self- Reported ADHD Symptoms is Associated with Sub-optimal and Inconsistent Temporal Decision Making’’.
- What are the exclusion criteria? More details should be specified in the method section. Because the age range of the sample is very wide, neurological disorders such as dementia and Alzheimer's disease can be seen in advanced ages. As it is known, these can cause deterioration in executive functions such as attention, decision making and impulsivity. If these are done, they should be specified in the method. If not, it should be added to the discussion as a limitation.
Author Response
Reviewer #3
Comment 1: In the discussion, it was stated as a limitation that ADHD could not be diagnosed clinically. However, the title is still very specious. Changing the title is recommended as ‘’Self- Reported ADHD Symptoms is Associated with Sub-optimal and Inconsistent Temporal Decision Making’’.
Response 1: We are grateful to the referee for picking up this inconsistency. The title was revised to state that ADHD symptoms were associated with sub-optimal and inconsistent decision-making. However, due to another comment, we removed the abbreviation from the title, and therefore it became very long, so we decided not to include "self-reported" in the revised title. Nevertheless, we agree it is very important to state that the symptoms are self-reported, and therefore it is clearly stated in the abstract.
Comment 2: What are the exclusion criteria? More details should be specified in the method section. Because the age range of the sample is very wide, neurological disorders such as dementia and Alzheimer's disease can be seen in advanced ages. As it is known, these can cause deterioration in executive functions such as attention, decision making and impulsivity. If these are done, they should be specified in the method. If not, it should be added to the discussion as a limitation.
Response 2: This is an important comment, the exclusion criteria appear in the data analysis under the methods section ("Two participants who failed the attention check and 23 who completed the whole study in less than four minutes were excluded from further analysis... The final sample consisted of 175 participants"), and this limitation was added to the discussion (paragraph 8: " old age was not considered an exclusion criterion, and therefore the age range was wide (18-74). Importantly, our findings could hence be attributed to the advanced age of some of our participants and the related neurological diseases and deficits. However, re-examining the data with age as a covariate did not have an impact on the results. We further made sure that participants did not report taking any medication for the treatment of neurological diseases. Future studies should take that issue into account in advance.").
Round 2
Reviewer 2 Report
Thank you.
Reviewer 3 Report
Accept in present form